# Gait and Stability Analysis of People After Osteoporotic Spinal Fractures Treated with Minimally Invasive Surgery

**DOI:** 10.3390/jfmk10040481

**Published:** 2025-12-17

**Authors:** Szymon Kaczor, Michalina Blazkiewicz, Malgorzata Kowalska, Adam Hermanowicz, Ewa Matuszczak, Justyna Zielińska-Turek, Justyna Hermanowicz

**Affiliations:** 1Neurosurgery Department, Military Hospital, 19-300 Elk, Poland; kaczorszymon@gmail.com; 2Chair of Physiotherapy Fundamentals, The Józef Piłsudski University of Physical Education, 00-968 Warsaw, Poland; michalina.blazkiewicz@awf.edu.pl; 3Pediatric Surgery and Urology Department, Medical University of Bialystok, 15-274 Bialystok, Poland; ahermanowicz@wp.pl (A.H.); ewa.matuszczak@umb.edu.pl (E.M.); 4Department of Neurology, National Medical Institute of the Ministry of Interior and Administration, 02-507 Warsaw, Poland; jzturek@gmail.com; 5Pharmacodynamics Department, Medical University of Bialystok, 15-089 Bialystok, Poland; justyna.hermanowicz@umb.edu.pl

**Keywords:** osteoporotic spinal fracture, postural stability, surgical treatment, minimally invasive surgery, fall risk

## Abstract

**Background:** Osteoporotic vertebral fractures in the thoracic–lumbar spine are common in older adults and can lead to pain, kyphotic posture, impaired postural control, and altered gait. These changes increase the risk of falls and reduce functional mobility, highlighting the need for effective assessment and intervention strategies. **Objectives:** To analyze stability and gait in patients who sustained a thoracic–lumbar spinal fracture and underwent minimally invasive surgery. **Methods:** Seventeen patients participated in this study (women = 11, age 68.36 ± 6.15 years, body weight 68.18 ± 12.8 kg, height 161.45 ± 5.26 cm; men = 6, age 62.67 ± 4.41 years, body weight 78.5 ± 20.36 kg, height 176.67 ± 12.64 cm). All participants had undergone minimally invasive spinal surgery using percutaneous screws reinforced with bone cement 12 months prior. Each patient underwent two assessments: postural stability measurement and biomechanical gait analysis. Statistical analysis was performed using Statistica software (StatSoft, PL), with significance set at *p* < 0.05. **Results:** In the stability test, seven participants could not complete the measurement due to falls (FRT = 6.45 ± 2.43), six performed within the normal range (FRT = 2.41 ± 0.9), and four were below the normal range for their age group (FRT = 2.22 ± 1.7). Patients exhibited slower walking speed, shorter stride length, and reduced hip extension during the stance phase (approximately 5° less) due to a forward-leaning posture and cautious gait. Foot placement was flat rather than heel-first, likely as a compensatory strategy to enhance safety. **Conclusions:** Patients after osteoporotic thoracic–lumbar vertebral fractures treated with minimally invasive surgery demonstrate shorter, wider, and slower steps, along with reduced postural stability, indicating a persistent risk of forward falls.

## 1. Introduction

Human gait is a highly complex motor activity that relies on the coordinated interaction of neural, musculoskeletal, and sensory systems. Its control depends on the integration of supraspinal centers, spinal pattern generators, proprioceptive feedback, vestibular input, and visual information, all of which contribute to maintaining balance, coordinating limb movements, and ensuring efficient locomotion [1,2]. Disruptions within this neural network—such as impaired sensory feedback, altered postural control, or reduced muscle activation—can lead to gait instability and increase the risk of falls. Understanding these neural mechanisms is essential to appreciate how pathological conditions affecting the spine may negatively influence gait.

Osteoporosis is a highly prevalent systemic skeletal condition, particularly among older adults. Epidemiological data indicate that osteoporotic vertebral fractures rank among the most common fractures in the elderly population, with nearly 500,000 new cases reported annually in Europe [3]. Compression fractures of vertebrae, the most frequent osteoporotic complication, often lead to back pain, kyphotic posture, reduced trunk muscle strength, and impaired sensorimotor control, all of which contribute to gait abnormalities [3,4,5]. Beyond structural weakening of bone, vertebral deformities may alter neural function by compressing nerve roots and disturbing proprioceptive pathways [6].

The mechanisms by which osteoporosis affects patients extend beyond bone fragility. Vertebral fractures may lead to persistent nociceptive signaling, impaired reflex responses, reduced postural stability, and compensatory gait patterns that increase mechanical strain on adjacent structures [6]. Treatment strategies must, therefore, be selected carefully to avoid further deterioration. Conservative approaches typically include pain management, physical therapy, and exercise, sometimes combined with nerve root blocks [4]. Surgical interventions—such as vertebroplasty, kyphoplasty, or minimally invasive stabilization using percutaneous screws with bone cement—are reserved for patients with persistent pain or neurological compromise [5]. Importantly, not all individuals qualify for such procedures; contraindications may include severe instability, infection, or comorbidities that increase surgical risk.

Given the importance of gait for functional independence, an objective and comprehensive assessment of locomotion is essential. Traditional clinical evaluations often rely on examiner experience and subjective assessment, limiting reproducibility and precision [7]. Modern biomechanical analyses address these limitations by evaluating kinematic parameters (e.g., spatio-temporal characteristics, joint angles, and linear and angular velocities) and kinetic parameters (e.g., ground reaction forces, plantar pressure distribution, and muscle moments) [7,8,9,10]. These measurements allow for a detailed assessment of gait quality and postural stability.

Despite the growing literature on gait biomechanics, evidence regarding functional outcomes in patients after minimally invasive treatment of thoracolumbar osteoporotic fractures remains limited. This study aims to fill this gap by assessing gait and stability in individuals who sustained thoracolumbar vertebral fractures and underwent minimally invasive surgery. By providing objective biomechanical data, this work contributes to the understanding of functional recovery in this population and may inform future rehabilitation strategies [6,7,10].

## 2. Materials and Methods

### 2.1. Study Group Characteristics

This prospective study was conducted at the Central Research Laboratory of the University of Physical Education in Warsaw. The Senate Committee of the Ethics of Research of the University of Physical Education in Warsaw approved this study (No. 84/PB/206). Before starting the measurements, the subjects were familiarized with the scheme, purpose, and course of this study, as well as being informed about the possibility of withdrawing from participation in the study at any time. A condition for joining this study was the written consent to participate in the experiment, which was completed by all respondents. The inclusion criteria were as follows: a spinal fracture in the thoracic–lumbar section; minimally invasive surgery 12 months prior to taking part in this study. The exclusion criterion was the use of different surgical methods. Seventeen people (11 women, W; and 6 men, M) after osteoporotic spinal fractures participated in the measurements, which are presented in Appendix A. Each patient had a spinal fracture in the thoracic–lumbar section and underwent minimally invasive surgery 12 months prior taking part in this survey. The operation was performed using the cutaneous method with perforated screws reinforced with bone cement [11]. We did not perform sample size calculations prior to this study.

### 2.2. Description of Biomechanical Test Methods

The measurements were taken using the equipment available at the Central Research Laboratory of the University of Physical Education in Warsaw. Each person underwent two types of measurements. The first measured postural stability, while the second was a biomechanical gait assessment.

#### 2.2.1. Stability Analysis

Two protocols and two devices were used in the patient stability measurements. Protocol 1 contained a fall risk test, and protocol 2 contained a standardized stability assessment.

Protocol 1. In the first step, fall risk while standing with both feet on the Biodex Balance System SD platform (Biodex, Shirley, NY, USA) was examined, which enabled assessment of the patient’s neuromuscular control in a closed kinematic chain.

The platform dimensions are 76 × 112 × 20 cm, and its diagonal is 55 cm. Platform tilt is possible by 20 degrees from the horizontal in all directions. The device includes twelve stability levels—plus a locked level for static testing—starting with level twelve, the most stable, and ending with level one, the least stable.

The same measurement procedure was performed for each patient. At the examiner’s instruction, the subject stood barefoot on the platform in a natural upright position with the upper limbs lowered along the body, eyes directed forward, without leaning the body forward or backward. During the test, the subject’s feet could not lose contact with the platform or change their initial position. The subject’s task was to stand still. The monitor feedback was enabled so that the subject could see the movement of the indicator informing about sway (Appendix A). During measurement, the subject did not communicate verbally with the examiner. The fall risk assessment test was performed only once with three measurements. The duration of each measurement was 15 s. The pause time was 5 s between each attempt. The obtained results are the average of three trials. The analyzed parameter is the fall risk test (FRT).

Protocol 2. The second measurement protocol analyzed the transfer of the center of pressure (CoP) of the feet on the ground. This was measured using the AMTI AccuSway platform (Advanced Mechanical Technology Inc., Watertown, MA, USA). The subject underwent stability testing while standing on two lower limbs (20 s each test), and while standing on the non-dominant lower limb (10 s each test). Standing with both feet consisted of the following tests: standing with open eyes (OE), with closed eyes (CE), and with closed eyes and a dual task (CE_DT). The same variants were performed when standing on one lower limb (non-dominant—left in all subjects). The aim of the dual task was to provide a distraction through completing the multiplication table.

#### 2.2.2. Gait Analysis

This study used the three-dimensional motion analysis system, Vicon Motion Capture Systems Ltd. (Oxford, UK). The Vicon system installed at the Central Research Laboratory consists of nine cameras with a sampling frequency set at 100 Hz, mounted around a 10-m walkway, which record the infrared light reflected from the markers. The system is synchronized with three Kistler piezoelectric platforms: two 9287BA models and one 9281B (900 × 600 × 210 mm) mounted on the walkway along with EMG test equipment (Noraxon, Scottsdale, AZ, USA). Each of the above-mentioned platforms is three-dimensional, and the high sensitivity is enabled by the piezoelectric transducers—four in each corner of the platform. Thus, it is possible to measure the three components of the ground reaction forces and to determine the coordinates of the center of pressure of the feet on the ground. Power, synchronization, and data transfer to the main computer were provided by the MxGiganet device. In the central unit, the collected data were subjected to specialist processing using the Vicon Nexus 2.14 program.

Before starting the study, each participant was subjected to anthropometric measurement, and the measured values were entered into the Vicon system. Then, 34 spherical passive markers with a diameter of 14 mm, covered with reflective material, were attached to the subject’s body. Markers were attached at the appropriate anatomical points of the body according to the Plug-In-Gait scheme. The actual gait test was preceded by a static test, performed to precisely determine the distances between the markers, to identify the markers, and to verify their completeness. The test subject was asked to remain still in the anatomical position for the duration of the test.

During the actual test, the subject had the task of walking (natural gait) along the measurement walkway. The actual recording is one in which each of the lower limbs was once on the platform, and there were no random errors, e.g., marker falling off, unnatural gait. Usually, each patient completed three natural gait trials, and one of these was selected for analysis. For further analysis, the following parameters were taken into account: spatio-temporal parameters (step length, gait velocity, single step time), the change curves in the angles, and muscle moments in the gait cycle. All the above-mentioned parameters were automatically calculated by the Vicon system. The above data obtained in the patient group were compared with the normative data of the Central Research Laboratory of the University of Physical Education in Warsaw.

### 2.3. Statistical Analysis

The statistical analyses began with an assessment of data distribution using the Shapiro–Wilk test (*p* < 0.05). This evaluation confirmed that all variables deviated from a normal distribution, which justified the use of non-parametric statistical procedures. As a result, the data are presented as medians with interquartile ranges.

Because the measurements were taken from dependent samples and the distribution of variables was non-normal, differences between conditions were analyzed using Wilcoxon’s signed-rank test. In addition to hypothesis testing, corresponding effect sizes (e.g., rank-biserial correlation) were calculated to quantify the magnitude of observed differences and enhance interpretability beyond *p*-values.

All statistical analyses were carried out using Statistica software 13.3 (StatSoft PL, Cracow, Poland), and statistical significance was set at *p* < 0.05.

## 3. Results

### 3.1. Stability Analysis—Protocol 1

For the FRT, the Biodex Balance System SD platform manufacturer prepared norms for particular age ranges (Appendix A). The results of the test, conducted on the seventeen people participating in this study, showed that seven people did not finish the measurement because they fell (FRT = 6.45 ± 2.43); six were in the normal range (FRT = 2.41 ± 0.9); four were below normal for their age group (FRT = 2.22 ± 1.7).

### 3.2. Stability Analysis—Protocol 2

After the Shapiro–Wilk test was performed, we found that the variables did not have a normal distribution (*p* < 0.05). It is worth noting that all the subjects performed the combinations of tests while standing with both feet. Therefore, below is an analysis of these results only. Statistical analysis with Wilcoxon’s test for dependent samples showed that there are statistically significant differences for the path length of the center of pressure of the feet on the ground when standing with eyes open and eyes closed (*p* = 0.0010), and for the CoP path length when standing with eyes open and eyes closed for the dual task (*p* = 0.0002). As shown, adding the dual task (distraction) to the eyes-closed test did not significantly change the stability test result compared with eyes open. We also found that all subjects had twice the maximum ranges of sagittal sway in each trial (Appendix A), indicating a physiological result. Only eight people performed the tests of standing on one lower limb when standing with eyes open. They obtained a mean path length score of 67.36 ± 29.95 cm. Only one person underwent the test with eyes closed, while no one performed the task of standing with eyes closed and the dual task.

### 3.3. Gait Analysis

Spatio-temporal parameters

The differences in mean parameters collected during this study are provided in Appendix A. The average walking speed was around 120 steps per minute, and based on the data in Appendix A, there are differences that indicate that people with osteoporotic spinal fractures move more slowly than healthy people. In the group of patients, the subjects took shorter steps by more than half.

Kinematic and kinetic parameters

The gait pattern of patients differs from that of healthy individuals. To compare the curves, we found the extreme points in the stance phase (0–60% of the gait cycle) and in the swing phase (60–100% of the gait cycle). When analyzing the kinematic curves, we can see that for the angle in the hip joint in the stance phase, two extreme points are visible—the maximum and the minimum—while in the swing phase, we have one maximum. For the knee joint, both in the swing phase and in the stance phase, two maximums are critical. For the ankle joint, two minimal values are critical in the stance and swing phases (Appendix A).

The data in Appendix A indicate that the patients are unable to perform the hip extension movement in the stance phase (−3.7 ± 14.1)° vs. (−8.7 ± 5.8)°, they have five degrees less extension, which is caused by the bent-over position and a more careful stride. There are two typical extremes (flexion) in the knee joint—less flexion in the stance phase by 15° and less flexion in the swing phase by 12°. In the ankle joint, the greatest differences can be seen for plantar flexion in the swing phase (−2.3 ± 2.5)° vs. (−19.8 ± 6.9)°. Such large 17° differences result from the fact that most likely the patients move the foot flat so that it can be safely placed on the ground, not starting from the heel. The above analysis of kinematic parameters is also confirmed for kinetic parameters, i.e., for muscle moments.

In the studied group, the ankle joint lacks the typical dorsiflexion, which causes patients to start their gait by putting their foot flat on the ground. In the knee joint, in our patients, we noted a completely different course of the average trajectories of the muscle moments in the stance phase. Such a result indicates impaired extensor function of the knee joint, characteristic of this group of patients, persisting in the stance phase (Appendix A).

## 4. Discussion

The primary objective of this study was to assess gait stability and postural control in patients who sustained thoracolumbar osteoporotic vertebral fractures and underwent minimally invasive surgical treatment. By providing objective biomechanical data on gait parameters and muscle moment patterns, our study aimed to address gaps in the literature regarding functional recovery in this population.

Gait is one of the most fundamental yet complex motor activities that humans perform subconsciously, and it is estimated that mastering it takes roughly the first seven years of life [1]. It is defined as a cyclical motor pattern involving the repeated coordination of the lower limbs and other body segments, characterized by rhythmic loss and recovery of balance across alternating stance and swing phases with minimal energy expenditure [1,2]. Because gait is essential for functional independence, its evaluation should be objective and comprehensive; however, traditional clinical assessments—often influenced by examiner experience—tend to lack precision and reproducibility. Modern biomechanical methods help overcome these limitations by quantifying key kinematic and kinetic parameters, offering a detailed insight into gait quality and postural control [7].

Osteoporotic vertebral fractures are among the most common fractures in elderly individuals, with nearly 500,000 new cases diagnosed annually in Europe [3]. While most fractures heal within a few weeks with pain resolution [12], up to 30% of patients experience prolonged pain and persistent functional deficits [3,5,13]. These deficits often include reduced postural control, altered center of mass, and increased risk of falls [3,4,5,6]. Falls among elderly people with osteoporosis can be the cause of morbidity and mortality [3,5,12,14]. The forward curvature of the trunk due to thoracolumbar hyperkyphosis further disrupts postural stability, leading to gait disturbances and compensatory strategies [6,9,15].

The neural and musculoskeletal mechanisms underlying these gait alterations are complex. Vertebral fractures and kyphotic deformities alter proprioceptive feedback and muscle activation patterns, resulting in decreased trunk extension, limited hip range of motion, and compensatory changes in knee and ankle function [6,9,16,17]. In our study, patients displayed reduced hip extension during stance, flatter foot placement, and abnormal ankle dorsiflexion patterns, indicating modifications in motor strategies likely aimed at minimizing instability and pain. Similarly, muscle moment trajectories in the knee and hip joints deviated from those of healthy individuals, reflecting increased effort by flexor muscles to maintain balance during stance [17]. These findings align with the current understanding of gait control and postural compensation following osteoporotic vertebral fractures.

Comparing our results with previous studies, the gait of patients after vertebral fractures—whether treated conservatively or surgically—remains slower and more cautious than in healthy individuals [18,19,20]. Stride length, cadence, and walking velocity are typically reduced, as observed in our cohort, with an average of approximately 120 steps per minute. Stability assessments showed that only a minority of patients could perform single-leg stance tasks, particularly with eyes closed or under dual-task conditions, highlighting persistent deficits in postural control despite surgical intervention [17,21].

Our study contributes to the field by providing detailed kinematic and kinetic analyses of gait in patients after minimally invasive thoracolumbar fracture repair, addressing a gap in the literature regarding functional outcomes post-surgery. While previous research largely focused on conservative treatment [21] or general clinical outcomes, our work offers precise biomechanical insights into lower-limb joint function, muscle moments, and balance strategies, which can inform targeted rehabilitation protocols and fall-prevention strategies.

Recent evidence suggests that outcomes after vertebral augmentation (vertebrplasty/kyphoplasty) are influenced not only by pain relief and radiologic correction, but also by the surgical technique, such as cement distribution, vertebral-height restoration, and post-operative spinal alignment—co-factors that may affect long-term mechanical stability and functional recovery [22,23,24]. Meta-analyses published in 2025 report a substantial rate of refracture (or new vertebral fractures) after minimally invasive treatment, especially in the presence of cement leakage or suboptimal cement filling, which may undermine spinal stability and increase the risk of gait and postural dysfunctions over time [25,26]. These findings reinforce the importance of close follow-up, anti-osteoporotic treatment, and possibly tailored rehabilitation in patients treated surgically for osteoporotic vertebral fractures.

### Study Limitations

The limitations of this study include a relatively small sample size, which is a consequence of restrictive inclusion criteria (patients with thoracolumbar spinal fracture who underwent minimally invasive surgery 12 months prior). Additionally, the cross-sectional design prevents the evaluation of long-term recovery trajectories. While objective gait and stability parameters were assessed, we did not include detailed evaluations of pain, muscle strength, or functional outcomes that may influence gait. Despite these limitations, this study provides valuable insight into post-surgical gait biomechanics and postural control in this specific clinical population, laying the groundwork for future longitudinal and larger-scale investigations.

## 5. Conclusions

Patients after thoracolumbar osteoporotic vertebral fractures treated with minimally invasive surgery exhibit altered gait with shorter, wider strides and reduced postural stability, indicating a persistent risk of forward falls. These findings directly address this study’s objective and emphasize the need for targeted post-operative rehabilitation. Future research should focus on longitudinal follow-up, larger cohorts, and interventions aimed at improving gait, balance, and fall prevention in this population.

## Data Availability

The original contributions presented in this study are included in the article/Appendix A. Further inquiries can be directed to the corresponding author.

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
