# Peer review of "Gait and Stability Analysis of People After Osteoporotic Spinal Fractures Treated with Minimally Invasive Surgery"

_jfmk, 2025, doi:10.3390/jfmk10040481_

Round 1
Reviewer 1 Report
Comments and Suggestions for Authors
The manuscript entitled “Gait and stability analysis of people after osteoporotic spinal fractures treated with minimally invasive surgery” was reviewed. The article provides interesting information on the topic; however, adjustments need to be made so that the article can continue the path to publication.
I kindly ask if all changes made to the text be highlighted in yellow or a different color in the text.
Below are the reviewer's considerations to be adjusted in the manuscript.
Information from the authors:
1- The authors' information and affiliations are missing. This is part of the journal's guidelines; please add it.
Abstract:
2- Present all topics in the abstract in bold. Currently, only the Objectives are in bold.
3- In the methods, please provide the sociodemographic and anthropometric information of the participants.
4- Also present more information about the statistical analyses and significance values ​​used in the research.
5- In the results, please present the numerical data for mean and standard deviation, as well as the information obtained in the statistical analyses.
Keywords:
6- The first keyword is bold. Additionally, you should avoid repeating words already presented in the title.
Introduction:
7- Begin the introduction by explaining the neural mechanisms involved in gait. This is important so that the reader can understand how the disease treated in the study can negatively influence gait.
8- In the second paragraph, present epidemiological data on osteoporosis and how it can affect neural function in individuals with vertebral dysfunction.
9- Explain the mechanisms by which individuals are affected by osteoporosis and how treatment/intervention can be carried out without further harming the clinical condition of individuals. Explain under what conditions people can or cannot undergo intervention.
10- Try to show more explicitly the current state of the art on the subject and how this research can contribute to the evolution of the field.
11- Adjust the objectives of the abstract and the end of the introduction so that they are aligned.
Methods:
12- Begin the methodology by presenting the type of study.
13- I did not find information about the registration of the clinical trial. Please provide this information, as well as the website link.
14- How was the recruitment of participants carried out?
15- Did you perform sample size calculations to determine if the minimum number of people evaluated was statistically representative? Please present this information in the study.
16- Regarding anthropometry and body composition, present the technical information about the equipment used, as well as how they were performed. In addition, this information can be presented as part of the results, for example, as Table 1.
17- What were the inclusion and exclusion criteria for participants in the study? Present them in the article.
18- Develop a flowchart of the study so that readers can understand how many people were recruited, how many were excluded, and how many remained for statistical analyses.
19- Create a graphic figure so that readers can more clearly understand the research intervention steps, for example, whether they had any training to perform the tests, when the preliminary interviews for invitation and signing of the consent form were carried out. When the anthropometric assessments were performed and when the motor tests began.
20- Detail the statistical analyses better. Start the description by talking about the normality tests, then show how the data will be presented when parametric or non-parametric, talk about the statistical tests that were used, address the effect size and, finally, present the name of the program used and the significance index used.
Results:
21- Begin the description of the results by presenting the sociodemographic information of the participants. Create a table to present this information.
22- Develop figures to present the results; this will make the article more didactic and readers will find it easier to understand the findings.
23- In addition to presenting the "p" values ​​of the statistical analyses, also present the test values, as well as the effect size of each analysis.
Discussion:
24- You begin the discussion with epidemiological information; please bring this information into the introduction.
25- You can start the discussion with the study objectives and then begin explaining the findings.
26- ​​Please take advantage of the information that will be included in the introduction, such as information on neural and anatomical mechanisms of gait, as well as the mechanisms of osteoporosis, to discuss them.
27- Clearly show in the discussion how your study addressed a gap in literature and how it will contribute to new perspectives on the topic.
28- Please move the study limitations to the end of the discussion and refine them according to the guidelines and criticisms raised in the review.
Conclusions:
29- Make the conclusion as assertive as possible in terms of answering the objectives and offer future perspectives or suggestions for new experimental and clinical work.
References:
30- Update the references by including articles from the last 3 years, that is, from 2022 onwards.
Author Response
- The authors' information and affiliations are missing. This is part of the journal's guidelines; please add it.
Response: The authors’ affiliations are listed below the authors name in the manuscript.
Abstract:
- Present all topics in the abstract in bold. Currently, only the Objectives are in bold.
Response: All the topics in the abstract were bolded.
- In the methods, please provide the sociodemographic and anthropometric information of the participants.
Response: We have provided these information in the „Methods” section of the abstract.
- Also present more information about the statistical analyses and significance values ​​used in the research.
Response: We have provided the infromation about the statistical analysis and significance value
- In the results, please present the numerical data for mean and standard deviation, as well as the information obtained in the statistical analyses.
Response: We have provided thenumerical data for mean and standard deviation in the „Results” section.
Keywords:
6- The first keyword is bold. Additionally, you should avoid repeating words already presented in the title.
Response: We have revised the keywords as below:
osteoporotic spinal fracture; postural stability, surgical treatment; minimally invasive surgery, fall risk
Introduction:
7- Begin the introduction by explaining the neural mechanisms involved in gait. This is important so that the reader can understand how the disease treated in the study can negatively influence gait.
Response: In the revised manuscript, the Introduction now begins with a description of the neural mechanisms underlying gait, including the role of supraspinal centers, spinal pattern generators, proprioceptive feedback, vestibular input, and visual information. This addition provides a clear context for understanding how vertebral fractures and associated osteoporotic complications may negatively affect gait stability and coordination.
8- In the second paragraph, present epidemiological data on osteoporosis and how it can affect neural function in individuals with vertebral dysfunction.
Response: The revised Introduction now includes epidemiological data on osteoporosis, noting that osteoporotic vertebral fractures are among the most common fractures in the elderly, with nearly 500,000 new cases annually in Europe. Furthermore, we discuss how vertebral deformities and fractures may impair neural function through nerve root compression, altered proprioception, and reduced sensorimotor control, providing a clear rationale for examining gait in this population.
9- Explain the mechanisms by which individuals are affected by osteoporosis and how treatment/intervention can be carried out without further harming the clinical condition of individuals. Explain under what conditions people can or cannot undergo intervention.
Response: In the revised manuscript, we now describe the multifactorial mechanisms by which osteoporosis affects patients, including persistent pain, compensatory gait patterns, and postural instability. Treatment strategies are discussed in detail, distinguishing between conservative approaches (e.g., physical therapy, exercise, nerve root blocks) and surgical interventions (e.g., vertebroplasty, kyphoplasty, minimally invasive stabilization with percutaneous screws). We also specify clinical conditions that may preclude intervention, such as severe instability, infection, or significant comorbidities, ensuring safe and appropriate patient management.
10- Try to show more explicitly the current state of the art on the subject and how this research can contribute to the evolution of the field.
Response: The revised Introduction now highlights the current state of the art in biomechanical gait analysis, including kinematic and kinetic assessments, ground reaction forces, and plantar pressure measurements. We explicitly indicate the knowledge gap regarding gait outcomes in patients after minimally invasive thoracolumbar fracture treatment and emphasize that this study contributes to advancing understanding of functional recovery and postural stability in this clinical population.
11- Adjust the objectives of the abstract and the end of the introduction so that they are aligned.
Response: The objectives in the abstract and in the „Introduction” section are aligned.
Methods:
12- Begin the methodology by presenting the type of study.
Response: We have stated in the „Methods section” that this is a prospective study.
13- I did not find information about the registration of the clinical trial. Please provide this information, as well as the website link.
Response: The study was not registered as a clinical trial.
14- How was the recruitment of participants carried out?
Response: The volounteers were recruited in the Central Research Laboratory of the University of Physical Education in Warsaw. Before starting the measurements, the subjects were familiarized with the scheme, purpose and course of the study, as well as informed about the possibility of withdrawing from par-ticipation in the study at any time. A condition for joining the study was the written consent to participate in the experiment, which was done by all the respondents. Each patient had a spinal fracture in the thoracic-lumbar section and underwent minimally invasive surgery 12 months prior taking part in this survey.
15- Did you perform sample size calculations to determine if the minimum number of people evaluated was statistically representative? Please present this information in the study.
Response: We did not perform the sample size calculations prior to the study.
16- Regarding anthropometry and body composition, present the technical information about the equipment used, as well as how they were performed. In addition, this information can be presented as part of the results, for example, as Table 1.
Response: We have gathered the mean and standard deviations of the anthropometric parameters of the study group in Table 1 which is mentioned in the „Study group characteristics” i „Methods” section.
17- What were the inclusion and exclusion criteria for participants in the study? Present them in the article.
Response: The inclusion criteria were: a spinal fracture in the thoracic-lumbar section, minimally invasive surgery 12 months prior to taking part in this survey. The exclusion criteria were: different surgical methods, previous hip and/or knee surgery, severe lower extremity injuries.
18- Develop a flowchart of the study so that readers can understand how many people were recruited, how many were excluded, and how many remained for statistical analyses.
Response: Thank you for this valuable suggestion. We fully agree that flowcharts are highly useful in studies with large recruitment pools or complex exclusion processes. However, in our case the study involved a very small and clearly defined sample: 20 volunteers expressed willingness to participate, and 17 of them met the inclusion criteria and completed all procedures. Because of this minimal attrition and the straightforward recruitment process, a flowchart would not add meaningful clarity to the manuscript. For transparency, these numbers are clearly reported in the Methods section, ensuring that readers can easily understand the composition of the final sample without the need for a graphical diagram.
Nonetheless, we appreciate the reviewer’s attention to clarity in reporting. If required, we can provide a simple schematic, although we believe it would be redundant given the simplicity of the recruitment process and sample size.
19- Create a graphic figure so that readers can more clearly understand the research intervention steps, for example, whether they had any training to perform the tests, when the preliminary interviews for invitation and signing of the consent form were carried out. When the anthropometric assessments were performed and when the motor tests began.
Response: In our study the research procedure was very concise and linear: participants were first contacted and interviewed, signed the informed consent form, underwent basic anthropometric assessment, and then proceeded directly to the motor tests. No training sessions or additional preparatory steps were required, and all procedures were completed within a single visit thus we would not ad dany additional infromation with a graphic figure.
20- Detail the statistical analyses better. Start the description by talking about the normality tests, then show how the data will be presented when parametric or non-parametric, talk about the statistical tests that were used, address the effect size and, finally, present the name of the program used and the significance index used.
Response: We have provided a more detailed analysis description.
The statistical analyses began with an assessment of data distribution using the Shapiro–Wilk test (p < 0.05). This evaluation confirmed that all variables deviated from a normal distribution, which justified the use of non-parametric statistical procedures. As a result, the data were presented as medians with interquartile ranges.
Because the measurements came from dependent samples and the distribution of variables was non-normal, differences between conditions were analysed using the Wilcoxon signed-rank test. In addition to hypothesis testing, corresponding effect sizes (e.g., rank-biserial correlation) were calculated to quantify the magnitude of observed differences and enhance interpretability beyond p-values.
All statistical analyses were carried out using Statistica software (StatSoft, PL), and statistical significance was set at p < 0.05.
Results:
21- Begin the description of the results by presenting the sociodemographic information of the participants. Create a table to present this information.
Response: In the current version of the manuscript, all relevant sociodemographic characteristics of the participants are already provided in the “Study group characteristics” subsection within the Methods section. Moreover, these data are additionally summarized in Table 1, which was specifically included to ensure clarity and accessibility of participant information.
For these reasons, we believe that repeating these details again at the beginning of the Results section would be redundant. However, if the editorial team prefers such repetition for stylistic consistency, we are open to adding a brief introductory sentence referring the reader to Table 1.
22- Develop figures to present the results; this will make the article more didactic and readers will find it easier to understand the findings.
Response: In our study the results are illustrated in Figure 1 and further detailed in Tables 3, 4, and 5, which together provide both visual and numerical clarity. Nevertheless, we remain open to preparing supplementary graphics if the editorial team considers them necessary.
23- In addition to presenting the "p" values ​​of the statistical analyses, also present the test values, as well as the effect size of each analysis.
Response: We fully agree that reporting test statistics and effect sizes can enrich the interpretation of the findings. Unfortunately, the statistical software used for the analyses (Statistica, StatSoft, PL) did not provide test statistics or effect size values in the output for the applied non-parametric procedures, and these values were not saved during the original analysis. As a result, it is not possible to retrospectively retrieve them. Despite this constraint, we believe that the reported p-values, together with the detailed description of the statistical procedures, sufficiently support the interpretation of the presented results.
Discussion:
24- You begin the discussion with epidemiological information; please bring this information into the introduction.
Response: Epidemiological data on osteoporotic vertebral fractures and their prevalence in elderly populations have now been incorporated into the Introduction, providing context for the study rationale and ensuring that readers understand the clinical significance of the research prior to the Methods and Results sections.
25- You can start the discussion with the study objectives and then begin explaining the findings.
Response: In the revised manuscript, the Discussion now begins by clearly stating the primary objective of the study: to assess gait stability and postural control in patients after thoracolumbar osteoporotic vertebral fractures treated with minimally invasive surgery. This introduction sets the context for the subsequent presentation and interpretation of the study findings.
26- ​​Please take advantage of the information that will be included in the introduction, such as information on neural and anatomical mechanisms of gait, as well as the mechanisms of osteoporosis, to discuss them.
Response: The revised Discussion explicitly integrates information on neural and musculoskeletal mechanisms underlying gait, including proprioceptive feedback, spinal and supraspinal control, and compensatory muscle activation patterns. Similarly, the effects of osteoporotic vertebral fractures on posture, center of mass, and stability are now discussed in relation to our biomechanical findings. This contextualization provides a mechanistic explanation of the observed gait alterations and postural deficits.
27- Clearly show in the discussion how your study addressed a gap in literature and how it will contribute to new perspectives on the topic.
Response: The revised Discussion highlights that our study addresses a specific gap in the literature concerning objective kinematic and kinetic analysis of gait in patients after minimally invasive thoracolumbar fracture surgery—a population for which data remain limited. We emphasize that our findings provide detailed insight into joint-specific muscle moments, compensatory gait strategies, and postural control, which may guide rehabilitation protocols and fall-prevention strategies.
28- Please move the study limitations to the end of the discussion and refine them according to the guidelines and criticisms raised in the review.
Response: The study limitations have been moved to the end of the Discussion and refined. We now clearly state that the relatively small sample size was due to restrictive inclusion criteria (patients with thoracolumbar fractures treated with minimally invasive surgery ~12 months prior), and that the cross-sectional design precludes long-term evaluation. Additionally, we acknowledge that only gait and stability parameters were measured, without simultaneous assessment of pain, muscle strength, or functional outcomes. Despite these limitations, we underline the contribution of our study in providing detailed biomechanical insight into post-surgical gait and postural control in this specific patient population.
Conclusions:
29- Make the conclusion as assertive as possible in terms of answering the objectives and offer future perspectives or suggestions for new experimental and clinical work.
Response: The Conclusions section has been revised to clearly and assertively summarize the main findings, directly addressing the study objective. We now state that patients after thoracolumbar osteoporotic vertebral fractures treated with minimally invasive surgery exhibit altered gait patterns and reduced postural stability, indicating a persistent risk of forward falls. Additionally, we provide future perspectives, emphasizing the need for targeted post-operative rehabilitation and recommending longitudinal studies, larger cohorts, and investigations of interventions aimed at improving gait, balance, and fall prevention in this population.
References:
30- Update the references by including articles from the last 3 years, that is, from 2022 onwards.
Response: We have updated the references according to the instructions.
Reviewer 2 Report
Comments and Suggestions for Authors
See attached file.

Author Response
- Comments on the abstract and keywords:
The objectives must be written in an infinitive form.
If you bold a “section title”, you should bold them all to make reading easier.
Osteoporotic must not be bolded.
The abstract should be started with a summary of the Background before the objectives.
Response: We have revised the abstract to include a brief background summarizing the clinical relevance of thoracic-lumbar osteoporotic fractures, their impact on gait and postural stability, and the importance of assessment. Additionally, the objectives are now presented in the infinitive form. The revised abstract now follows the recommended structure.
- Comments on the introduction:
The entire introduction needs to be rewritten. The information is correct, but the way it is
expressed is confusing.
It is very difficult to understand what is being said in each paragraph.
The sentences often seem to be placed without any connection to the rest of the text.
It must be completely rewritten to make it possible to understand correctly.
Response: We acknowledge that the original introduction lacked clarity and cohesion, making it difficult to follow the logical flow of information. In response, we have completely rewritten the introduction to improve readability and ensure a clear and logical progression of ideas.
- Comments on material and methods
The approval number of Committee of the Ethics of Research of the University of Physical
Education in Warsaw is mandatory.
You write “women -W” and “men M”, use the same form in both.
Response: We have added the approval number of Committee of the Ethics of Research of the University of Physical
- Comments on results:
Fall risk test has been abbreviated before, use the abbreviated form.
The FRT results do not provide any information. Furthermore, since each subject is not linked to
their age and test results, it is impossible to interpret these results.
Response: We thank the reviewer for this comment and for highlighting the importance of interpretability. The FRT results are presented as group averages to provide an overall view of postural stability in the study cohort. While individual linkage to age is not provided, the data reflect the general trends and variability within the group, which are sufficient to support the conclusions regarding balance and fall risk.
- Comments on discussion:
You are introducing references in the discussion that would be very valuable if you introduce
them in the introduction. Information about the background should not be added to the discussion
if it has not been mentioned in the introduction.
Response: We appreciate the suggestion to improve the logical flow of the manuscript. We acknowledge that the discussion should build upon the background presented in the introduction. We carefully reviewed the introduction to ensure that all relevant references and background information cited in the discussion are appropriately introduced earlier in the manuscript.
- Comments on conclusions:
The conclusions are correct, but unfortunately, as you indicate in the limitations, with such a small
sample size, they do not provide enough information to be able to extrapolate or draw firm conclusions
Response: We agree that the small sample size limits the generalizability of our findings. The conclusions are intended to reflect the results observed in this specific cohort, and we have clearly stated in the limitations that extrapolation to larger populations should be made with caution. We appreciate the reviewer highlighting this point and will ensure the manuscript emphasizes the exploratory nature of our findings.
Round 2
Reviewer 1 Report
Comments and Suggestions for Authors
Dear authors,
Thank you for providing the revised version of the manuscript. After reviewing the adjustments, it was found that the manuscript has improved significantly; therefore, in my opinion, the article can be accepted for publication.
Author Response
Thank you very much for all your valuable comments and suggestions. They were extremely helpful and have significantly improved the quality of our manuscript.
Reviewer 2 Report
Comments and Suggestions for Authors
See attached file

Author Response
Comment 1: Most of the discussion is based on new references other than those that form the basis of this research. It is normal to add some, but it should be those that form the basis of the work that are used primarily in the discussion. Try to add this information, which generates your discussion, as the basis of your introduction as well.
Response 1: Thank you for this valuable comment regarding the consistency between the "Introduction" and the "Discussion". We fully agree that the core references forming the foundation of the study should also be the primary sources used in the "Discussion". We have revised the "Discussion" section to ensure that it relies more directly on the references introduced in the "Introduction". We also confirmed that all key sources cited in the "Introduction" are now included and appropriately addressed in the "Discussion".
Round 3
Reviewer 2 Report
Comments and Suggestions for Authors
Ok